# The Effectiveness of Pressure Safety Valves in Chemical Supply Systems to Prevent Fire, Explosion, and Overpressure in the Korean Semiconductor Industry

**Kyeong-Seok Oh [1,*], Euittum Jeong [2], Woo Sub Shim [3,*] and Jong-Bae Baek [1]**

[1] Department of Safety Engineering, Korea National University of Transportation, Chungju 27469, Republic of Korea

[2] Department of Chemical Engineering, Chonnam National University, Gwangju 61186, Republic of Korea

[3] Chemical Accident Prevention Division, Occupational Safety and Health Headquarter, Ministry of Employment and Labor, Sejong 30117, Republic of Korea

\* Correspondence: happyguyoks@hanmail.net (K.-S.O.); shimws0720@korea.kr (W.S.S.); Tel.: +82-10-7621-1323 (K.-S.O.); +82-10-6391-2052 (W.S.S.)

**Abstract:** This study was conducted to review the safety and appropriateness of PSV (Pressure Safety Valve) installation in the supply tank, which is a pressure vessel included in supply systems dedicated to supplying the acid/alkaline substances used in the Korean semiconductor manufacturing process. Three aspects of design, risk assessment, and regulations were reviewed to determine if there is a source of pressure higher than the maximum allowable working pressure (MAWP) of the supply tank that could cause fires, explosions, and overpressure. In the case of the design review, all 17 overpressure scenarios described in API Standard 521, i.e., pressure-relieving and depressuring systems, were reviewed, and there was no overpressure scenario above the maximum allowable working pressure (MAWP). Then, the risk assessment, i.e., the Hazard and Operability Study (HAZOP) technique, was used, and as a result of reviewing all possible risk situations, we can state that there were no overpressure scenarios that can exceed the design pressure of the supply tank; thus, we decided that the installation of a PSV on top of the supply tank is unnecessary. Finally, accident prevention measures against overpressure, such as the KS B 6750-3 system design and the Korean Industrial Standard, were reviewed from a legal point of view. It was confirmed that the hazardous chemical supply system for the semiconductor industry designed in this study has several protective functions to prevent fires, explosions, and overpressure. As a result of reviewing the above three aspects, it can be said that there is no need to install a pressure safety valve in a pressure vessel storing hazardous chemicals.

**Keywords:** pressure safety valve (PSV); pressure vessel; supply system; fire; explosion; overpressure; risk assessment; Hazard and Operability Study (HAZOP); semiconductor industry

## 1. Introduction

The Ministry of Trade, Industry and Energy in Korea announced on 13 April 2023 that Korea's exports and imports of information and communications technology (ICT) goods in the month of March was USD 15.8 billion (down 32.2 percent year-on-year) and USD 11.9 billion (down 7.9 percent), respectively. Trade balance stood at a surplus of USD 4.0 billion. Today, the electronics industry, represented by semiconductors, has grown rapidly in scale. In the case of semiconductors, they accounted for about 19% of Korea's exports in 2022 and became Korea's foremost exported product, both in a colloquial and genuine sense [1]. Typically, numerous substances are used for various purposes in the electronics industry. Most of these substances are classified as hazardous chemicals and are managed as regulated targets in Korea by the Industrial Safety and Health Act, the Chemical Substances Control Act, the Dangerous Substances Safety Management Act, and the High Pressure Gas Safety Management Act.

In the case of the semiconductor manufacturing process, dozens to hundreds of hazardous chemicals are used, and they are largely classified into chemicals, special gases, and precursors depending on the nature of the material and handling method. Among them, 'chemicals' are substances that are liquid at room temperature and pressure but can be divided into organic substances and inorganic substances according to their properties [2]. Inorganic substances are again classified into acids and alkalis and are stored and handled according to the characteristics of each substance. Storage facilities refer to facilities that store hazardous chemicals indoors, outdoors, or underground for the purpose of manufacturing, using, selling, and transporting hazardous chemicals. The chemicals are transferred to subsequent processes for multi-purpose use.

In particular, chemical processes involve risks such as flammability, explosiveness, and toxicity. For this reason, the identification of these hazards is very important to ensure the safe design and operation of these process plants [3]. That is why the number of technologies developed and utilized to prevent unsavory accidents in chemical processes from the Industrial Revolution to the present has increased. Compared to other chemical processes, the semiconductor manufacturing process differs slightly depending on the type of raw material, amount of handling, and type of reaction. Although semiconductor processing uses a small amount of chemicals compared to other chemical processes, there are many types and facilities throughout whole processes. In addition, special chemicals such as $SiH_4$ and $NF_3$, which are widely used in manufacturing processes, often cause fires or explosions and also have toxicities [4–6]. Therefore, in the semiconductor industry, risk assessment is essential in order to prevent large-scale accidents such as fires and explosions.

Currently, the most commonly used techniques for risk assessment are: Hazard and Operability Study (HAZOP), Failure Modes and Effects Analysis (FMEA), What If Analysis, Failure Modes, Effects and Criticality Analysis (FMECA), Process Hazard Analysis (PHA), Event Tree Analysis (ETA), Fault Tree Analysis (FTA), BOWTIE, BAYESIAN NETWORK, Hazard Identification (HAZID), and Layer Of Protection Analysis (LOPA) [7–12]. The HAZOP Study has been applied globally to address the risk analysis of plants where major chemical accidents may occur. It is considered an appropriate and important inspection used to assess the potential risk of malfunctioning equipment in terms of newly introduced processes or the restarting of existing processes. Even in the semiconductor industry, the HAZOP Study is already a commonly used method when risk analysis is required.

The chemical supply system is shown in Figure 1 and is divided into the following rank order according to the supply order: Auto Clean Quick Coupler (ACQC)—Transfer unit—Supply unit—Chemical Insert Box (CIB) -Valve Manifold Box (VMB)—FAB. The supply tank included in the supply unit has a design pressure of about 0.7 MPa and is classified as a pressure vessel according to the Occupational Safety and Health Act. As a result, safety certification needs to be obtained during the manufacturing process, and safety inspections have to be conducted periodically.

Pressure vessels such as supply tanks are equipped with various safety devices, including gauges to check internal conditions such as pressure gauges and thermometers and safety valves to relieve pressure in the case of overpressure [13–15]. A pressure vessel is a vessel designed to hold a gas or liquid at a pressure substantially different from the ambient pressure. Manufacturing methods and materials for pressure vessels can be selected for pressure applications and depend on vessel size, contents, operating pressure, mass constraints, and the number of items required. Pressure vessels can be hazardous due to the fact that they have their own energy and hazardous materials inside, and fatalities have occurred throughout their history of development and operation. As a result, the design, manufacturing, and operation of pressure vessels are regulated by legally supported technical authorities. For this reason, the definition of pressure vessel varies from country to country. In Korea, a pressure vessel is classified as a vessel with a design pressure of 0.2 MPa or more. Pressure safety valves should be installed in pressure vessels in accordance with Article 261 (installation of safety valves, etc.), Paragraph 1 of the Industrial Safety and Health Standards Act.

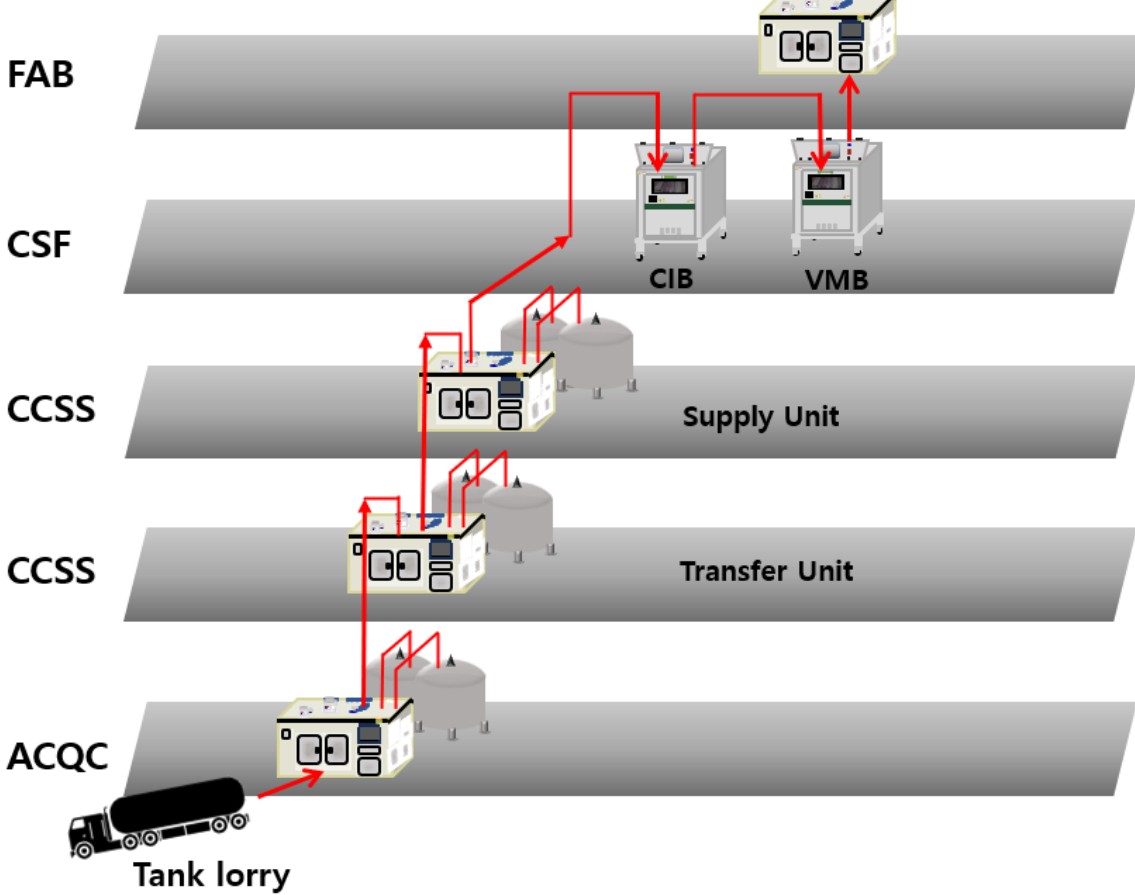

**Figure 1.** Chemical supply system in the semiconductor manufacturing process.

Pressure safety valves are considered to be the last safety device that prevents the leakage of hazardous substances and fires and explosions due to the damage of the pressure container when overpressure occurs as a result of an abnormal situation [16]. Failure of the pressure safety valve primarily causes equipment damage and can extend to process damage through accidents such as fires and explosions. Therefore, risk assessments must be thoroughly conducted, and the pressure safety valve's operating performance must be periodically inspected. A pressure safety valve is the most important safety device related to pressure. When the internal pressure in a facility or pressure vessel rises abnormally, the pressure in the facility or pressure vessel rises above the maximum allowable working pressure, and the pressure safety valve releases the material on the inside to the outside to prevent damage to the facility or pressure vessel. There are several types of pressure safety valves, including spring-type safety valves, rupture disk-type safety valves, melting, relief valves, and automatic pressure control devices. Among them, the product that must be inspected in a specific facility is the spring-type safety valve [17–24].

In addition to pressure vessels, safety valves shall be installed in positive displacement compressors, positive displacement pumps (with a shut-off valve installed on the discharge), and pipes (those that are blocked by two or more valves and may rupture due to thermal expansion of liquid at ambient temperature). In addition, since safety valves must also be installed in chemical facilities and auxiliary facilities that may exceed the maximum operating pressure for a facility, numerous safety valves are installed and managed in industrial sites.

In addition, a pressure safety valve installed according to the above standards must undergo a popping test every year. Popping tests are conducted to check whether the pressure safety valve operates properly at the set pressure, and they must be performed using a pressure gauge calibrated by the national calibration institution. If a rupture disk is

installed in front of the PSV, a popping test can only be conducted once every two years. If the plant obtained the highest grade in Process Safety Management (PSM) operation [25], a popping test is only necessary once every four years.

This study aims to identify all potential risks that may occur while operating a chemical supplying system and to judge whether pressure safety valves are really needed on a supply tank which is specified as a pressure vessel. This paper is organized into four sections: (1) Introduction, (2) Experimental Setup, (3) Results and Discussion, and (4) Conclusions.

## 2. Experimental Setup

The facility covered in this study is the supply tank described in the supply unit mentioned above, and chemicals are supplied from the supply tank to the CIB and VMB through nitrogen pressurization. Therefore, the supply tank was designed with a pressure higher than the nitrogen supply pressure (0.7 MPa vs. 0.5 MPa) and corresponded to a pressure vessel. As shown in Figure 2, two pressure safety valves and two rupture disks were installed in parallel on the upper part of the supply tank, and one pressure safety valve was installed separately in the nitrogen supply line to relieve overpressure caused by the failure of the regulator. Regarding the storage tank, nitrogen pressure transfers chemicals from the tank lorry to the storage tank. At this time, we planned for the storage tank to be excluded from the subject of this review because the storage tank was equipped with a vent line that operates under atmospheric pressure. Next, the storage tank uses a magnetic pump to transport the chemical to the supply tank. When transferring to the supply tank with a magnetic pump, auto valve 1 is opened via the interlock, and the vent line is opened. Subsequent, auto valve 2 closes and enters the same state as the normal pressure storage tank. After being transferred to the supply tank, the chemical is transferred from the supply tank to CIB and VMB via nitrogen pressurization. During nitrogen pressurization, auto valve 1 is closed via the interlock to close the vent line, and auto valve 2 is opened to generate pressure in the supply tank. Therefore, the supply tank becomes a pressure vessel, thereby warranting its inclusion in this review.

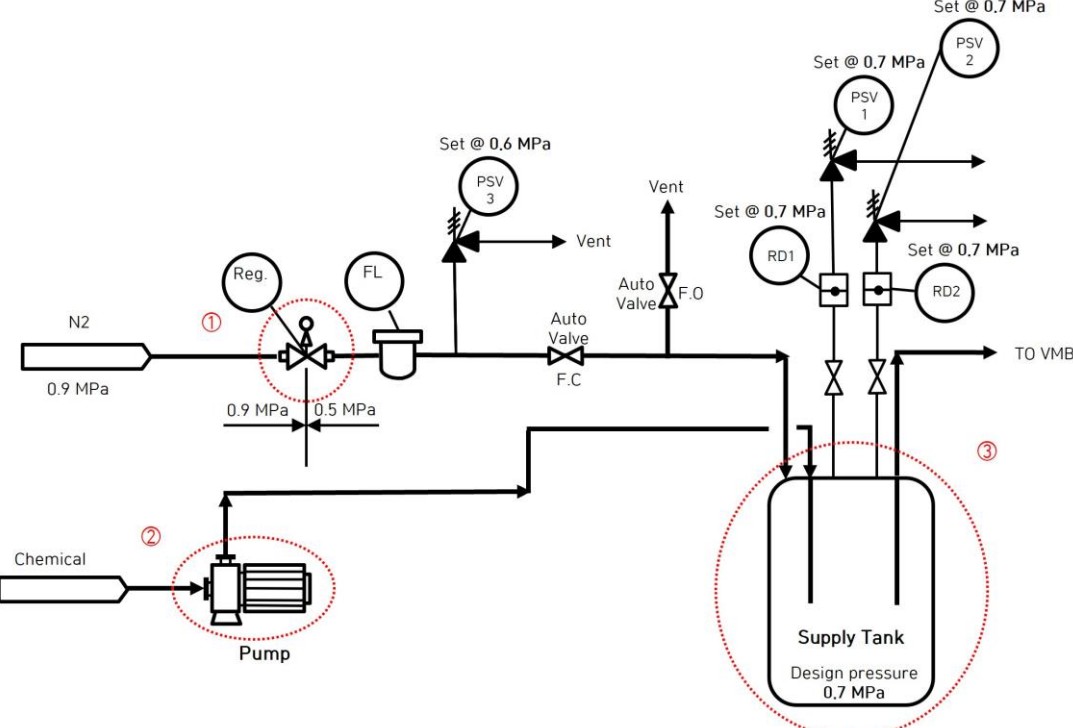

**Figure 2.** The configuration of the chemical supply system.

The required capacity of the facility's pressure safety valves should be calculated according to the possible overpressure scenario. In addition, a safety valve with a higher rated capacity than the largest required capacity scenario should be installed. In case of an overpressure scenario, 17 overpressure possibilities were reviewed based on the API Standard 521. In other words, depending on the facility, the applicable overpressure scenario may be just one scenario or several [26,27].

In the electronics industry, there are cases where the overpressure scenario is not clear. This is because external fire scenarios can be considered in the case of supply tanks handling organic chemicals [28]. However, it is not appropriate to consider external fire scenarios in the case of inorganic chemicals such as acids/alkalis. Here, an external fire is a scenario in which a liquid surface fire is generated due to the leakage of organic substances and heat is applied to the pressure vessel. This is because inorganic chemicals such as acids/alkalis are incombustible substances. In general, in the electronics industry, rooms are divided according to the properties of chemicals, and no chemicals other than the corresponding chemicals exist in each room.

In this study, the effectiveness of the safety valve was reviewed in three aspects: Design review, risk assessment, and regulatory review for the acid/alkali chemical supply system. For the design review, all 17 overpressure scenarios described in API Standard 521 were reviewed. For risk assessment, all potential risks were reviewed using the HAZOP technique based on PIDs (Piping & Instrument Diagrams). Lastly, in the review of the regulations, the content of the Korean Industrial Standard KS B 6750-3, which is delegated by Article 84 of the Occupational Safety and Health Act, was reviewed.

## 3. Results and Discussion

This section can be divided into subsections. We will provide a concise and accurate description of the experimental results, our interpretation of them, and the experimental conclusions that can be drawn from them.

In this study, all scenarios presented in API 521 were reviewed to determine whether or not the overpressure protection by the system design mentioned in KS B 6750-3 was applicable. A risk assessment (HAZOP) was also performed based on design data to confirm that there was no overpressure-generating source.

### 3.1. Design and Application Review

Overpressure means that the design pressure is exceeded due to an increase in system pressure due to thermal imbalance, excessive pump flow, and other causes. Therefore, an analysis of the cause and magnitude of overpressure is necessary to maintain a safe process system. In general, we can extract all pressure scenarios for a facility according to relevant Process Flow Diagrams (PFDs), Piping & Instrument Diagrams (PIDs), cause and effect diagrams, control loops, specified fire zones, operating conditions, and so forth. Hence, a PSV on a facility can be subject to single-case overpressures, such as blocked outlets and control valve failures, or to common cases, such as instrument air failure and cooling water failure.

The sources that can cause overpressure in the acid supply system are ① to ③ (shown in Figure 2): ①—$N_2$ regulator failure (MPR), ②—closed outlet on storage tank while magnetic pump is operated, and ③—external fire. First, in the case of nitrogen regulator failure (MPR), the system pressure cannot exceed 0.6 MPa by PSV3, so tank overpressure does not occur. Second, in the case of a closed outlet in the storage tank while magnetic pumps are being operated, there is no overpressure due to the fact that the supply tank design pressure is higher than the pump shut-off pressure. Third, in the case of external fires, since it is impossible to form a liquid pool fire due to the chemical characteristics of acid/alkali, overpressure due to an external fire does not occur. Accordingly, this case is excluded from consideration. The results of the 17 scenarios reviewed for the supply tank are shown in Table 1. All commonly reviewed overpressure scenarios were reviewed,

but scenarios in which overpressure exceeds the design pressure and maximum allowable working pressure of the supply tank were not reviewed.

**Table 1.** Overpressure scenarios as per API STD 521.

| No. | Overpressure Scenario | Overpressure Possibility |
|---|---|---|
| 1 | Closed Outlet<br>(1) When the supply tank outlet valves are closed, magnetic pump shut-off pressure can affect the supply tank, but there is no overpressure due to the supply tank design pressure being higher than the pump shut-off pressure.<br>(2) When the supply tank outlet valves are closed and the N2 regulator fails to open, maximum N2 pressure can affect the supply tank, but there is no overpressure due to the PSV (set pressure: 0.6 MPa) installed in the N2 supply line. | Not possible |
| 2 | Cooling-water failure to condenser<br>There is no condenser in this system; no overpressure occurred. | N/A |
| 3 | Top-tower reflux failure<br>There is no top-tower reflux in this system; no overpressure occurred. | N/A |
| 4 | Sidestream reflux failure<br>There is no sidestream reflux in this system; no overpressure occurred. | N/A |
| 5 | Lean-oil failure to absorber<br>There is no lean-oil in this system; no overpressure occurred. | N/A |
| 6 | Accumulation of noncondensables<br>There are no noncondensables in this system; no overpressure occurred. | N/A |
| 7 | Entrance of highly volatile material<br>There is no highly volatile material in this system; no overpressure occurred. | N/A |
| 8 | Overfilling<br>The supply tank can become overfilled by the magnetic pump, but there is no overpressure due to the supply tank design pressure being higher than the magnetic pump shut-off pressure. | Not possible |
| 9 | Failure of automatic controls<br>(1) $N_2$ regulator fails to open in the $N_2$ supply line.<br>➜ Refer to 1–2.<br>(2) $N_2$ regulator fails to close in the $N_2$ supply line.<br>➜ No overpressure due to $N_2$ supply stoppage.<br>(3) Auto valve fails to open in the $N_2$ supply line.<br>➜ No overpressure due to the limited inlet pressure (0.5 MPa) of the auto valve.<br>(4) Auto valve fails to close in the $N_2$ supply line.<br>➜ No overpressure due to $N_2$ supply stoppage.<br>(5) Auto valve fails to open in the vent line.<br>➜ No overpressure due to $N_2$ vent stoppage.<br>(6) Auto valve fails to close in the vent line.<br>➜ No overpressure due to limited $N_2$ pressure (0.5 MPa).<br>(7) Auto valve fails to open in the $H_2SO_4$ supply line.<br>➜ No overpressure due to there being no valve block out.<br>(8) Auto valve fails to close in the $H_2SO_4$ supply line.<br>➜ No overpressure due to limited $N_2$ pressure due to the PSV (0.6 MPa).<br>(9) Auto valve fails to open in the inlet/outlet of the magnetic pump.<br>➜ No overpressure due to there being no valve block out.<br>(10) Auto valve fails to close in the inlet/outlet of the magnetic pump.<br>➜ No overpressure due to the supply stoppage by the pump. | Not possible |

**Table 1.** *Cont.*

| No. | Overpressure Scenario | Overpressure Possibility |
|---|---|---|
| 10 | Abnormal process heat or vapor input<br>    Even if the auto valve in the $N_2$ line is open, there is a PSV (0.6 MPa) on the $N_2$ line.<br>Then, there is no overpressure in the supply tank (0.7 MPa). | Not possible |
| 11 | Internal explosions or transient pressure surges<br>    There are no explosion or transient pressure surges sources in this system; no overpressure occurred. | N/A |
| 12 | Chemical reaction<br>    There is no chemical reaction in this system; no overpressure occurred. | N/A |
| 13 | Hydraulic expansion<br>    There is no hydraulic expansion in this system; no overpressure occurred. | N/A |
| 14 | External fire<br>    External fire is not applicable for acid/alkali chemicals; no overpressure occurred. | N/A |
| 15 | Heat transfer equipment failure<br>    There is no heat transfer equipment in this system; no overpressure occurred. | N/A |
| 16 | Power failure (steam, electric, or other)<br>(1)    If the instrument air fails, all auto valve positions will be moved to the fail position (fail open or fail close). Then, there is no overpressure.<br>(2)    If the electric power fails, the magnetic pump will stop, and the $H_2SO_4$ supply will be stopped. Then, there is no overpressure. | Not possible |
| 17 | Maintenance<br>    There are no overpressure sources by maintenance in this system; no overpressure occurred. | N/A |

In the case of a closed outlet, there is no possibility of overpressure. When the supply tank outlet valves are closed, the magnetic pump shut-off pressure can affect the supply tank, but there is no overpressure due to the supply tank design pressure being higher than the pump shut-off pressure. Also, when the supply tank outlet valves are closed and the $N_2$ regulator fails to open, the maximum $N_2$ pressure can affect the supply tank, but there is no overpressure due to the PSV (set pressure: 0.6 MPa) being installed in the $N_2$ supply line.

In the case of overfilling, there is no possibility of overpressure. The supply tank can be overfilled by a magnetic pump, but there is no overpressure due to the supply tank design pressure being higher than the magnetic pump shut-off pressure. In the case of a failure of automatic controls, there is no possibility of overpressure. Neglectful valve operation by workers performing the process can cause the valve position to be the reverse of the normal operating conditions. This position is mainly caused by human error and can be avoided by performing the process according to standard operating procedures. Additionally, open or closed errors in control valves are caused by electronic or mechanical signal errors. This error usually only affects one valve at a time and needs to be analyzed on a valve-by-position, situation-by-case basis.

In cases of a failure of the abnormal process heat or vapor input, there is no possibility of overpressure. This failure is caused by (1) increased supply of heating medium such as fuel oil or fuel gas to the fired heater; (2) heat transfer from new and clean heat exchangers after retrofitting; (3) the fuel supply control valve not fully opening; (4) the supply pressure of the heating steam changing from the normal range to the maximum pressure. In addition, abnormal steam inflow can be caused by the full opening of the upstream control valve or failure via the upstream relief or inadvertent opening of the valve. Problems may arise if the required venting capacity is not equal to or greater than the vapor accumulation expected under venting conditions. However, there is no heat input in this system; thus, it is not possible. In the cases of power failure (i.e., steam, electric, or other), there is no possibility of overpressure. If the instrument air fails, all auto valve positions will be moved to a fail position (fail open or fail close), and then there will be no overpressure. Also, if the electric power fails, the magnetic pump will stop, and the $H_2SO_4$ supply will be stopped. Then, there will be no overpressure. This situation can be considered a failure of process

controllers such as programmable logic controllers and distributed control systems. It is necessary to separate the potential impact of the failure of all control loops from the case where one loop fails but all other loops remain active. In general, the required venting capacity should be greater than the steam produced by the system's heat build-up to avoid problems. Excluding the 5 scenarios listed above, 12 overpressure scenarios were not applicable.

*3.2. Risk Assessment*

Through conducting a risk assessment of the system, we examined whether there is a scenario in which the safety valve installed on the supply tank is operated. Our risk assessment was performed using the HAZOP (Hazard and Operability) technique. The HAZOP technique is a method of examining possible causes of deviation during normal process operations, predicting the results and risk levels, and then identifying appropriate safety measures. The HAZOP technique is one of the most used risk assessment techniques [29–34]. HAZOP research is an activity that helps identify and evaluate possible hazards to personnel, equipment, or processes and helps prevent accidents. The HAZOP Team should investigate the causes and consequences that may occur (e.g., fires, explosions, and toxic material release) when the process exits its normal operating conditions. After identifying the causes and consequences, safety measures to prevent the occurrence of accidents should be identified. After the analysis at the corresponding analysis node is finished, the previous method is repeatedly executed at the next analysis node to expand the entire process. The HAZOP sheet is a measure used to mitigate risks for the operation and maintenance departments of the facility and serves as a guiding document that must be reviewed prior to performing the process.

Among the acid/alkali chemicals, the sulfuric acid supply device, which is the most used material in the electronics industry, was selected, and the results are shown in Figures 3–5 below. Risk is calculated as a combination of Likelihood and Severity. The criteria are shown in Table 2, and the definitions of each risk (according to their risk level) are as follows: Risk 1—Low-Risk (Acceptable), Risk 2—Medium-Risk (as low as reasonably practicable), Risk 3—High-Risk (additional action required); and Risk 4—Unacceptable Risk (need for a re-design). The detailed criteria of Likelihood and Severity are shown in Appendix A.

**Table 2.** 4 × 4 Risk Matrix Diagram (a combination of Likelihood and Severity).

| Likelihood \ Severity | 1 | 2 | 3 | 4 |
|---|---|---|---|---|
| 1 | 1 | 2 | 2 | 2 |
| 2 | 2 | 2 | 3 | 3 |
| 3 | 2 | 3 | 3 | 4 |
| 4 | 2 | 3 | 4 | 4 |

The first node analyzed was the $N_2$ line for the $H_2SO_4$ supply line, the second node analyzed was the $H_2SO_4$ supply line (to the end user), and the third node analyzed was the $H_2SO_4$ charge line (from the storage tank).

Through the risk assessment, all possible risks to the acid/alkali supply facility were reviewed. There was a "nitrogen supply regulator fail open case" in which an overpressure higher than the design pressure of the supply tank could occur, but this can be resolved through a safety valve installed in the nitrogen supply line. In accordance with the in-house risk comparison table and risk management standards, the risks identified through our risk assessment were analyzed and categorized as insignificant or minor risks, and it was decided that additional recommendations were not necessary. As a result of reviewing all possible scenarios with the help of the HAZOP worksheet, there were no overpressure scenarios that could exceed the design pressure of the supply tank, so the safety valve installed on the supply tank was deemed unnecessary.

**HAZOP Worksheet**

Node No. 1
Node Description: N$_2$ line for H$_2$SO$_4$ supply

| No. | Deviation | Causes | Consequence | L | S | R | Safeguards | ML | MS | MR | Recommendation |
|---|---|---|---|---|---|---|---|---|---|---|---|
| 1 | No/Less Flow | Filter plugging | Production issue due to insufficient H$_2$SO$_4$ supply | 1 | 2 | 2 | 1. PAL-1 on Supply tank<br>2. FT-01<br>3. PAL-53 (after chemical filter)<br>4. periodic replacement | 1 | 2 | 2 | |
| 2 | | AV-06 malfunction close (on N$_2$ line to H$_2$SO$_4$ supply tank) | Production issue due to insufficient H$_2$SO$_4$ supply | 2 | 2 | 2 | 1. PAL-1 on Supply tank<br>2. FT-01<br>3. PAL-53 (after chemical filter) | 1 | 2 | 2 | |
| 3 | | AV-07 malfunction close (on N$_2$ vent line) | Increase of Supply tank pressure leading to Supply tank damage and leakage of H$_2$SO$_4$, Potential operator injury | 2 | 3 | 3 | 1. Pump permissive operation (Pump starts when Supply tank pressure is lower than set pressure)<br>2. TK-A Design Pressure (0.7 MPa) > N$_2$ Operating Pressure (0.5 MPa)<br>3. PAH-1 on Supply tank | 1 | 3 | 2 | |
| 4 | More FLow | Regulator-01 fail open | Increase of Supply tank pressure over than Supply tank Design Pressure(0.7 MPa) leading to Supply tank damage and leakage of H$_2$SO$_4$, Potential operator injury | 2 | 3 | 3 | 1. PSV-03 on N$_2$ Line (SP : 0.6 MPa)<br>2. Regulator-02 (second regulator)<br>3. PAH-1 on Supply tank | 1 | 3 | 2 | |
| 5 | | Regulator-02 fail open | Minor H$_2$SO$_4$ supply issue due to pressure fluctuation | 2 | 2 | 2 | 1. PAH-1 on Supply tank | 1 | 2 | 2 | |
| 6 | | AV-06 malfunction open (on N$_2$ line to H$_2$SO$_4$ supply tank) | No significant consequence | | | | | | | | |
| 7 | | AV-07 malfunction open (on N$_2$ vent line) | Production issue due to insufficient H$_2$SO$_4$ supply | 2 | 2 | 2 | 1. PAL-1 on Supply tank<br>2. FT-01<br>3. PAL-53 (after chemical filter) | 1 | 2 | 2 | |
| 8 | Low Pres. | refer to Flow | | | | | | | | | |
| 9 | High Pres. | refer to Flow | | | | | | | | | |
| 10 | Low Temp. | N/A | | | | | | | | | |
| 11 | High Temp. | N/A | | | | | | | | | |
| 12 | Low Level | N/A | | | | | | | | | |
| 13 | High Level | N/A | | | | | | | | | |
| 14 | etc. | N/A | | | | | | | | | |

**Figure 3.** The results of the HAZOP Study_Node 1 (N$_2$ line for the H$_2$SO$_4$ supply).

**HAZOP Worksheet**

Node No. 2
Node Description: H$_2$SO$_4$ Supply Line (to end user)

| No. | Deviation | Causes | Consequence | L | S | R | Safeguards | ML | MS | MR | Recommendation |
|---|---|---|---|---|---|---|---|---|---|---|---|
| 1 | No/Less Flow | AV-04 malfunction close | Production issue due to insufficient H$_2$SO$_4$ supply | 2 | 2 | 2 | 1. PAL-51/52/53<br>2. FT-01 | 1 | 2 | 2 | |
| 2 | | FL-01 (Chemical Filter) plugging | No significant consequence (Serveral Filters are used) | | | | | | | | |
| 3 | | AV-21 malfunction close | Production issue due to insufficient H$_2$SO$_4$ supply | 2 | 2 | 2 | 1. PAL-53<br>2. FT-01 | 1 | 2 | 2 | |
| 4 | More Flow | AV-04 malfunction open (during Supply Tank A filling) | Minor H$_2$SO$_4$ supply issue due to pressure fluctuation | 2 | 2 | 2 | 1. PAL-51/52/53<br>2. FT-01 | 1 | 2 | 2 | |
| 5 | | AV-21 malfunction open | No significant consequence (normally open) | | | | | | | | |
| 7 | Low Pres. | refer to Flow | | | | | | | | | |
| 8 | High Pres. | N/A | | | | | | | | | |
| 9 | Low Temp. | N/A | | | | | | | | | |
| 10 | High Temp. | N/A | | | | | | | | | |
| 11 | Low Level | N/A | | | | | | | | | |
| 12 | High Level | N/A | | | | | | | | | |
| 13 | etc. | N/A | | | | | | | | | |
| 14 | | | | | | | | | | | |
| 15 | | | | | | | | | | | |

**Figure 4.** The results of the HAZOP Study_Node 2 (H$_2$SO$_4$ supply line to the end user).

### 3.3. Legal Review

Pressure vessels with a design pressure of 0.2 MPa or more must receive safety certification from the Ministry of Employment and Labor in accordance with Paragraph 1 of Article 84 of the Industrial Safety and Health Act. Detailed standards for safety certification are mentioned in the Ministry of Employment and Labor Notice No. 2020-41 "Notice on Safety Certification for Hazardous Machines". In addition, Article 11 (Production and Safety Standards) of the same notice stipulates that the design and manufacturing standards of pressure vessels must comply with the Korean Industrial Standard [KS B 6750-3 (General Industrial Pressure Vessel)]. According to 10.1.15 "Overpressure protection by system design" of KS B 6750-3, it states that pressure relief devices are not required for

pressure vessels where the pressure is self-limiting. This pressure is less than the maximum allowable operating pressure (MAWP) of the pressure vessel at the set temperature and satisfies the following conditions [35]: (1) the above should be described in the manufacturer's data report, (2) the user should conduct a detailed analysis to identify and test all possible overpressure scenarios, and (3) the analysis results should be documented. Therefore, as a result of the above technical review, it can be said that the system's design provides overpressure protection.

**HAZOP Worksheet**

Node No. 3
Node Description: $H_2SO_4$ Charge line (from storage tank)

| No. | Deviation | Causes | Consequence | L | S | R | Safeguards | ML | MS | MR | Recommendation |
|---|---|---|---|---|---|---|---|---|---|---|---|
| 1 | | AV-08 malfunction close (during Storage Tank A filling) | Production issue due to insufficient $H_2SO_4$ supply | 2 | 2 | 2 | 1. FAL-06 | 1 | 2 | 2 | |
| 2 | | AV-05 malfunction close (during Storage Tank A filling) | Production issue due to insufficient $H_2SO_4$ supply | 2 | 2 | 2 | 1. FAL-06 | 1 | 2 | 2 | |
| 3 | No/Less Flow | | DP-01 Damage | 2 | 2 | 2 | 1. PAL-03 | 1 | 2 | 2 | |
| 4 | | DP-01 trip | Production issue due to insufficient $H_2SO_4$ supply | 2 | 2 | 2 | 1. PAL-03 2. Stand-by pump 3. Stand-by Supply tank | 1 | 2 | 2 | |
| 5 | | FL-03 plugging | No significant consequence (Serveral Filters are used) | | | | | | | | |
| 6 | More Flow | AV-08 malfunction open | No significant consequence | | | | | | | | |
| 7 | | AV-05 malfunction open | No significant consequence | | | | | | | | |
| 8 | Wrong Direction | AV-03 malfunction open (during Supply Tank B filling) | Minor production issue due to insufficient $H_2SO_4$ supply | 2 | 2 | 2 | | | | | |
| 9 | Low Pres. | refer to Flow | | | | | | | | | |
| 10 | High Pres. | N/A | | | | | | | | | |
| 11 | Low Temp. | N/A | | | | | | | | | |
| 12 | High Temp. | N/A | | | | | | | | | |
| 13 | Low Level | N/A | | | | | | | | | |
| 14 | High Level | N/A | | | | | | | | | |
| 15 | etc. | N/A | | | | | | | | | |

**Figure 5.** The results of the HAZOP Study_Node 3 ($H_2SO_4$ charge line (from the storage tank)).

## 4. Conclusions

In this study, the supply tank, which is a pressure container included in the acid/alkali supply system in the electronics industry, was reviewed with respect to three aspects, namely, design review, risk assessment, and legal review, to see if there was a pressure source higher than the Maximum Allowable Working Pressure (MAWP). Firstly, regarding the design review, all 17 overpressure scenarios described in *API STANDARD 521* were reviewed, and there was no overpressure scenario above the allowable working pressure (MAWP). Secondly, regarding risk assessment, after examining all of the risky situations that can occur using the HAZOP technique, there was no situation in which the safety valve installed on the top of the supply tank operated. Thirdly, regarding the aspect of reviewing the laws and regulations, reviewing overpressure protection according to the system design in KS B 6750-3 and the Korean industrial standard confirmed that the acid/alkali supply system in the electronics industry is protected from overpressure due to system design.

Therefore, as a result of reviewing these three aspects, the installation of safety valves can be excluded from the supply tank in the acid/alkali supply system of the semiconductor industry. Based on this study, if the government revises the relevant regulations, it is expected that the industry-handling pressure vessels will be very welcome as they can prevent risks or potentially excessive investments regarding the addition of unnecessary safety devices. In addition, since most safety valves are installed in high places, according to the relevant regulations, they should be inspected once a year, and at this time, a fall accident often occurs. However, if the relevant regulations are amended, safety accidents themselves can be eliminated at source. In addition, the results of this study are expected to contribute to the expansion of the industrial use of overpressure prevention systems through future studies such as QRAs (Quantitative Risk Assessments), FTA (Fault Tree Analysis), ETA (Event Tree Analysis), and on-site evaluations using PHAST (Process Hazard Analysis Tool). Finally, if the safety of the chemical process in the semiconductor industry

is guaranteed, the semiconductor industry will become a sustainable industry and will develop further.

**Author Contributions:** Conceptualization, K.-S.O.; methodology, J.-B.B.; validation, J.-B.B., E.J. and W.S.S.; writing—original draft preparation, K.-S.O. and W.S.S.; writing—review, K.-S.O. and W.S.S.; editing, W.S.S. All authors have read and agreed to the published version of the manuscript.

**Funding:** This work was partially supported by the 2022 Chemical Substance Specialization Graduate School Support Project of the "Chemical Safety Management Expert Training Project" (Government Business Number: B0080531001698) sponsored by the Korean Ministry of Environment.

**Institutional Review Board Statement:** Not applicable.

**Informed Consent Statement:** Not applicable.

**Data Availability Statement:** Some or all data used in this research are available from the corresponding author upon request.

**Conflicts of Interest:** The authors declare no conflict of interest.

## Abbreviations

| | |
|---|---|
| ACQC | Auto Clean Quick Coupler |
| CIB | Chemical Insert Box |
| ETA | Event Tree Analysis |
| FMEA | Failure Modes and Effects Analysis |
| FMECA | Failure Modes, Effects and Criticality Analysis |
| FTA | Fault Tree Analysis |
| HAZID | Hazard Identification |
| HAZOP | Hazard and Operability |
| LOPA | Layer of Protection Analysis |
| MAWP | Maximum Allowable Working Pressure |
| PHA | Process Hazard Analysis |
| PSV | Pressure Safety Valve |
| QRA | Quantitatively Risk Assessment |
| VMB | Valve Manifold Box |

## Appendix A. The Criteria for Likelihood and Severity

### Likelihood

Likelihood 1: Occurs less than once in 100 years
Likelihood 2: Occurs once in 10–100 years
Likelihood 3: Occurs once in 1–10 years
Likelihood 4: Occurs more than once a year

### Severity

| | Human Damage | Production Downtime | Production Loss |
|---|---|---|---|
| Severity 1 | No damage | No downtime | No loss |
| Severity 2 | Minor injuries | 1–2 days | Less than USD 1 million |
| Severity 3 | Serious injuries | 3–6 days | USD 1–10 million |
| Severity 4 | More than one fatality | More than a week | More than USD 10 million |

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
