# Peer review of "The Effectiveness of Pressure Safety Valves in Chemical Supply Systems to Prevent Fire, Explosion, and Overpressure in the Korean Semiconductor Industry"

_fire, doi:10.3390/fire6090344_

Round 1

Reviewer 1 Report

- there are several miscalculations in the risk score (L-2 x S-2 = R-2?)

Need clarification on the use of the risk matrix and the risk score.
- the risk matrix used
- the classification of low medium and high
- the scale of probability and severity

Reviewer 2 Report

Reviewer comments #: 

The main topic of the paper “Hazardous chemical supply system to prevent fire, explosion and overpressure: Korean Semiconductor Industry Case Study” is in line with research themes published within Fire Journal. Generally, the topic of the manuscript is interesting and the manuscript provided useful information but requires minor modification to be published. I have the following comments that the authors should implement in the revised manuscript before publication, please see the minor comments below: 

1. The abstract is fine but still need improvement and included more specific information about the main points from this study

2. In the section of introduction, the information mentioned by authors in the lines (55-75) should be supported by recent publications to support this information.

3. The authors should be included more recent publications in the introduction to enrich the introduction in more specific details. 

4. I suggest including the all abbreviations as a table at the end of the manuscript.

5. The section of Materials and Methods written in good way, but I recommend authors to give more explanation for the three aspects: Design review, risk assessment, and regulatory review.

6. The sentence of “In this study, all scenarios presented in API 521 were reviewed to determine whether or not the overpressure protection by the system design mentioned in KS B 6750-3 was applicable. The risk assessment (HAZOP) was also performed based on design data to confirm that there was no overpressure generating source. Therefore, as a result of the above technical review, it can be said that the system has overpressure protection by system design” should be move to the beginning of the section 3. Also, I suggest authors to move the last sentence of the section 3 “Therefore, as a result of the above technical review, it can be said that the system has overpressure protection by system design” to the conclusion section.

7. The submitted manuscript suffers from minor mistakes in writings and grammatically errors. Please go over the manuscript for other grammatical errors and correct all of them.

8. I suggest authors to reduce the conclusion section and including just the main points that concluded from this study. 

Thank you
